# Deciphering synergetic core-shell transformation from [Mo$_6$O$_{22}$@Ag$_{44}$] to [Mo$_8$O$_{28}$@Ag$_{50}$]

Zhi Wang [1], Hai-Feng Su[2], Chen-Ho Tung[1], Di Sun [1] & Lan-Sun Zheng[2]

The structural transformation of high-nuclearity silver clusters from one to another induced by specific stimuli is of scientific significance in terms of both cluster synthesis and reactivity. Herein, we report two silver-thiolate clusters, [Mo$_6$O$_{22}$@Ag$_{44}$] and [Mo$_8$O$_{28}$@Ag$_{50}$], which are templated by isopolymolybdates inside and covered by $^i$PrS$^-$ and PhCOO$^-$ ligands on the surfaces. Amazingly, the [Mo$_8$O$_{28}$@Ag$_{50}$] can be transformed from [Mo$_6$O$_{22}$@Ag$_{44}$] by adding PhCOOH which increases the degree of condensation of molybdates template from Mo$_6$O$_{22}$$^{8-}$ to Mo$_8$O$_{28}$$^{8-}$, then enlarging the outer silver shell from Ag$_{44}$ to Ag$_{50}$. The evolution of solution species revealed by time-dependent electrospray ionization mass spectrometry (ESI-MS) suggests a breakage-growth-reassembly (BGR) transformation mechanism. These results not only provide a combined assembly strategy (anion-template + induced transformation) for the synthesis of silver-thiolate clusters but also help us to better understand the complex transformation process underpinning the assembly system.

[1] Key Laboratory of the Colloid and Interface Chemistry, Ministry of Education, School of Chemistry and Chemical Engineering, Shandong University, 250100 Jinan, China. [2] State Key Laboratory for Physical Chemistry of Solid Surfaces, Department of Chemistry, College of Chemistry and Chemical Engineering, Xiamen University, 361005 Xiamen, China. These authors contributed equally: Zhi Wang, Hai-Feng Su. Correspondence and requests for materials should be addressed to D.S. (email: dsun@sdu.edu.cn)

With regard to their ubiquitous argentophilicity and variable coordination fashions[1–3], Ag(I) coordination complexes, especially for silver clusters, are gorgeous in structural diversity and physicochemical properties[4–6]. However, the synthesis of high-nuclearity silver clusters is always tedious and frankly a trial-and-error process. Overwhelming these synthesis barriers has promoted the appearance of exquisite assembly strategies including anion templation and geometric polyhedral principle[7,8], which have pushed the assembly of silver clusters to a higher level of sophistication[9,10]. From the known largest silver(I) solid cluster (Ag$_{490}$)[11] to the largest silver(I) cage (Ag$_{180}$)[12], we have witnessed the fruitful advances in this field. However, there is still vast room for improvement to realize the manipulation over such clusters at the molecular level, not just synthesizing them randomly. Inspired by the LEIST (ligand-exchange-induced size/structure transformation) methodology widely used in Au$_n$(SR)$_m$ nanoclusters[6], we would like to study whether the similar stories can be observed in their silver cousins. In 2012, Mak et. al., reported the reaction of a famous Cl@Ag$_{14}$ cluster with AgClO$_4$, which gave a larger Cl$_6$Ag$_6$@Ag$_{30}$ cluster, thus realizing the cluster enlargement[13]. Following this, a polyoxovanadate-templated Ag$_{30}$ cluster was subjected to acid/base stimulations, which only resulted in the reversible conversion of configurations of [V$^V_{10}$V$^{IV}_2$O$_{34}$]$^{10-}$ from $D_{3d}$ to $C_{2h}$, however, the silver shell was kept invariable[14]. These sporadic reports indicated the structural transformation of silver clusters inside and out by specific stimuli is still a challenging task.

In order to achieve structural transformation of silver clusters, two prerequisites are needed: (i) flexible silver shells and (ii) variable anion templates. Installing monocarboxylate ligands on the surface of silver clusters will endow the silver clusters some flexibility because the carboxylate belongs to hard base with respect to thiolate, thus forming relatively weak bonding with soft acid Ag(I) atoms[15]. As such, when the stimuli-induced post-reaction proceeds in solution, the carboxylates can partially disassociate then induce the rearrangement of surface silver atoms. For variable anion templates, polyoxometalates (POMs) are the best candidates due to their mutable forms depending on pH values[16].

With all above considerations in mind, herein we use thiol and benzoic acid as mixed ligands to construct a silver cluster, [Mo$_6$O$_{22}$@Ag$_{44}$($^i$PrS)$_{20}$(PhCOO)$_{16}$(CH$_3$CN)$_2$]·2CH$_3$CN (SD/Ag44; SD = SunDi), which can be transformed to another larger silver cluster, [Mo$_8$O$_{28}$@Ag$_{50}$($^i$PrS)$_{24}$(PhCOO)$_{18}$(CH$_3$CN)$_2$)]·4CH$_3$CN (SD/Ag50) by the reaction with additional PhCOOH. To the best of our knowledge, this is the first report of the PhCOOH-induced structural transformation simultaneously involving enlargements of inner anion template (Mo$_6$O$_{22}^{8-}$ → Mo$_8$O$_{28}^{8-}$) and outer silver shell (Ag$_{44}$ → Ag$_{50}$). Such transformation is also proved by the time-dependent ESI-MS of SD/Ag44 after adding PhCOOH and a breakage-growth-reassembly (BGR) mechanism is also proposed.

## Results

**Structures of SD/Ag44 and SD/Ag50.** The SD/Ag44 was synthesized by solvothermal reaction of polymeric ($^i$PrSAg)$_n$ precursor, PhCOOAg and ($^n$Bu$_4$N)$_4$(α-Mo$_8$O$_{26}$) in CH$_3$CN at 65 °C (Fig. 1). After the reaction, the yellow tufted crystals can be collected as the bulk sample of SD/Ag44. If adding another portion of PhCOOH (0.32 mmol) into above reaction mother liquor without removing crystals of SD/Ag44, then continuing to react again under the same condition will produce yellow block crystals of SD/Ag50.

The molecular structures of SD/Ag44 and SD/Ag50 were determined by single-crystal X-ray diffraction (SCXRD) analysis

(Supplementary Table 1). Both of them crystallize in triclinic $P$-1 space group with a complete cluster in the asymmetric unit. As shown in Fig. 2a, the overall structure of SD/Ag44 is a slightly squashed spheroid composed of 44 silver atoms and covered by 20 $^i$PrS$^-$, 16 PhCOO$^-$ ligands and two CH$_3$CN molecules. Interiorly, an unusual Mo$_6$O$_{22}^{8-}$ anion in situ generated from ($^n$Bu$_4$N)$_4$(α-Mo$_8$O$_{26}$) supports the outer Ag$_{44}$ shell. The diameter of SD/Ag44 is roughly 1.2 nm, if removing the organic shell. Based on their coordination modes, 20 $^i$PrS$^-$ ligands are divided into two types comprised of 2 μ$_3$ and 18 μ$_4$ (Fig. 2b), which capped on the irregular silver trigons, tetragons or pentagons with the Ag-S bond lengths spanning from 2.3641(17) to 2.7589(18) Å. The PhCOO$^-$ ligands with μ$_2$-κ$^1$:κ$^1$, μ$_3$-κ$^1$:κ$^2$, μ$_4$-κ$^2$:κ$^2$, and μ$_4$-κ$^1$:κ$^3$ modes cap on the edges or faces of silver polygons (Supplementary Fig. 2). The Ag-O$_{benzoate}$ bond lengths locate in the range of 2.114(5)−2.787(7) Å. Two additional CH$_3$CN molecules finally finish the coverage of Ag$_{44}$ shell (Ag-N: 2.338(7) and 2.395(6) Å), which is further consolidated by the Ag···Ag interactions (2.8272(7)−3.4289(8) Å)[17,18].

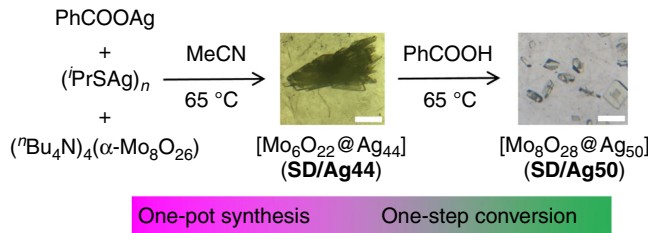

PhCOOAg
+
($^i$PrSAg)$_n$ —MeCN→ [Mo$_6$O$_{22}$@Ag$_{44}$] —PhCOOH→ [Mo$_8$O$_{28}$@Ag$_{50}$]
+                65 °C        (SD/Ag44)        65 °C        (SD/Ag50)
($^n$Bu$_4$N)$_4$(α-Mo$_8$O$_{26}$)

One-pot synthesis          One-step conversion

**Fig. 1** Synthesis and transformation routes for **SD/Ag44** and **SD/Ag50**. The scale bar is 1 mm

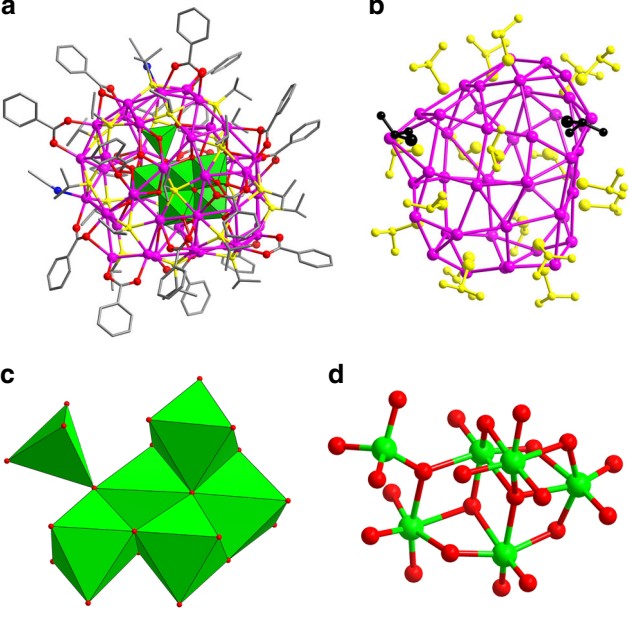

**Fig. 2** Single-crystal X-ray structure of **SD/Ag44**. **a** The molecular structure of **SD/Ag44**. The inner POM anion template is shown in polyhedral mode (Color legend: Ag: purple; S: yellow; Mo: green; O: red; C: gray; N: blue). **b** The distributions of $^i$PrS$^-$ ligands on the Ag$_{44}$ shell sorted by different coordination modes individually colored. Black: μ$_3$ and yellow: μ$_4$. The polyhedral (**c**) and ball-and-stick (**d**) modes showing the inner Mo$_6$O$_{22}^{8-}$ anion

Notably, a $Mo_6O_{22}^{8-}$ anion was trapped into the $Ag_{44}$ cluster and should be in situ transformed from $\alpha$-$Mo_8O_{26}^{4-}$. Bond-valence sum (BVS) calculations for six Mo atoms were performed and confirmed all of them are +6 oxidation state (Mo1-Mo6: 5.744, 5.455, 5.792, 5.531, 5.766, and 6.173)[19]. This indicates that no redox reaction occurs under the reaction condition. The $Mo_6O_{22}^{8-}$ in **SD/Ag44** is constructed from five edge-shared $MoO_6$ octahedra adding one $MoO_4$ tetrahedron by sharing one vertex (Fig. 2c). Alternatively, it can be seen as three $Mo_3O_4$ opened cubanes fused together by sharing two $Mo_2O_2$ rhombus faces with another vertex-shared $MoO_4$ outside (Fig. 2d). This $Mo_6O_{22}^{8-}$ has 14 terminal O atoms, 5 $\mu_2$ bridging O atoms and 3 $\mu_3$ bridging O atoms. The Mo-O bond distances are in the range of 1.730(5)−2.328(4) Å, suggesting unusual elongated characteristic of Mo-O bonds compared to other $Mo_6O_{22}^{8-}$ anions in classic POM chemistry[20]. Using the $\mu_1$ and $\mu_2$ bridging O atoms, $Mo_6O_{22}^{8-}$ binds total 30 Ag atoms around it with the Ag-$O_{POM}$ bond lengths ranging from 2.255(4) to 2.793(5) Å (Supplementary Fig. 3). To date, only three isomeric $Mo_6O_{22}^{8-}$ anion templates have been reported in silver clusters such as face-shared $Mo_4O_4$ double cubanes[21,22], two $Mo_3O_4$ opened cubanes sandwiching a $Mo_4O_4$ cubane by face-sharing[23], and three face-shared $Mo_3O_4$ opened cubanes[24], as summarized in Supplementary Table 2. By comparisons, the $Mo_6O_{22}^{8-}$ in **SD/Ag44** is the fourth isomer trapped by silver cluster. Moreover, such $Mo_6O_{22}^{8-}$ is also different from those observed in POM-based inorganic–organic hybrids (Supplementary Table 3)[25–27], suggesting a new $Mo_6O_{22}^{8-}$ structure.

When adding another portion of PhCOOH to the mixture after the synthesis of **SD/Ag44** for second step reaction, we can isolate a larger silver cluster, **SD/Ag50**. SCXRD analysis revealed that it is a larger cluster than **SD/Ag44**. The overall structure of **SD/Ag50** comprised of 50 Ag atoms, forming a nearly ball-like shape with a diameter of *ca.* 1.4 nm, which is co-protected by 24 $^iPrS^-$, 18 $PhCOO^-$ ligands, and two $CH_3CN$ molecules (Fig. 3a). The ratio of $^iPrS$/PhCOO in **SD/Ag50** (4:3) is slightly larger than that in **SD/Ag44** (5:4), indicating the re-organization of two kinds of ligands on the surface after structural transformation. As shown

in Fig. 3b, 24 $^iPrS^-$ ligands show $\mu_4$ and $\mu_5$ binding modes, whereas $PhCOO^-$ ligands show $\mu_2$-$\kappa^1$:$\kappa^1$, $\mu_3$-$\kappa^1$:$\kappa^2$, $\mu_4$-$\kappa^1$:$\kappa^3$, or $\mu_4$-$\kappa^2$:$\kappa^2$ modes. The Ag-S and Ag-$O_{benzoate}$ bond lengths fall in the ranges of 2.248(7)−2.917(4) and 2.238(13)−2.771(12) Å, respectively. Due to the lack of regular silver polygons on the surface, such $Ag_{50}$ cluster doesn't exhibit any geometrical polyhedron feature. The $Ag_{50}$ shell was further consolidated by rich Ag···Ag interactions falling in the range of 2.851(2)−3.431(2) Å.

This 50-nucleus cluster is also templated by a novel rod-like POM, formulated as $Mo_8O_{28}^{8-}$, which can be seen as three $Mo_4O_4$ cubanes fused together by sharing two $Mo_2O_2$ rhombus faces (Fig. 3c), or described to eight edge-shared $MoO_6$ octahedra (Fig. 3d). The BVS[19] also verified +6 oxidation state of eight Mo atoms (Mo1-Mo8: 5.865, 5.431, 5.648, 5.889, 5.868, 5.748, 6.092, and 5.738). The surface of $Mo_8O_{28}^{8-}$ has 20 terminal, 4 $\mu_2$ bridging and 4 $\mu_4$ bridging O atoms. Among these O atoms, only 20 terminal O atoms participate to the coordination with Ag atoms (Supplementary Fig. 4). Compared to the starting material $\alpha$-$Mo_8O_{26}^{4-}$, $Mo_8O_{28}^{8-}$ has higher negative charges and more O atoms, which endow it as a better template in the assembly of silver clusters. The $Mo_8O_{28}^{8-}$ in **SD/Ag50**, to the best of our knowledge, has never been reported in POM-templated silver clusters and classic POM chemistry.

Inspired by the above discussed structure features of **SD/Ag44**, both $PhCOO^-$ ligands and weakly coordinated $CH_3CN$ molecules are the potential reaction active sites, and the POM template can be also varied depending on the solution acidity. In fact, we found the transformation of **SD/Ag44** to **SD/Ag50** can facilely work by adding additional PhCOOH without changing any other reaction conditions. Based on the comparisons of their structures, such cluster-to-cluster structural transformation is a rare occurrence that simultaneously involved inner anion template ($Mo_6O_{22}^{8-} \rightarrow Mo_8O_{28}^{8-}$) and outer silver shell ($Ag_{44} \rightarrow Ag_{50}$). In this transformation process, the degree of condensation of molybdate was increased, resulting in the growth of $Mo_6O_{22}^{8-}$ to $Mo_8O_{28}^{8-}$. The larger POM template thus enlarges the outer silver shell from $Ag_{44}$ to $Ag_{50}$. We also tried to synthesize **SD/Ag50** without the transformation step by directly adding more PhCOOH into the reaction system, however, the **SD/Ag50** can be only isolated with a quite low yield (<5%), whereas using the transformation synthesis route, it can reach up to more than 50%. Thus, PhCOOH-induced structural transformation might provide a new method for the high-yield synthesis of some silver clusters that are otherwise difficult to access. Moreover, this transformation reaction should be a thermodynamics controlled reaction, so **SD/Ag50** should be more stable than **SD/Ag44**.

**ESI-MS of SD/Ag44 and SD/Ag50.** In order to investigate solution behaviors of both clusters, the ESI-MS of **SD/Ag44** and **SD/Ag50** dissolved in dichloromethane were measured in the positive ion mode. As shown in Fig. 4a, there are a series of +2 species (**1a**-**1k**) centered at the *m/z* range of 2000–8000. The most dominant peak centered at 4392.48 (**1e**) was assigned to [$\Delta$ $(^iPrS)_{26}(PhCOO)_8(CH_2Cl_2)_2(H_2O)$]$^{2+}$ based on the superimposable observed and simulated isotope patterns (Cal. 4392.49; $\Delta = Mo_6O_{22}@Ag_{44}$ hereafter). Similar formula assignment for each labeled species in ESI-MS was also performed as listed in Supplementary Table 4. Based on the assigned formula, we found the core of **SD/Ag44** is quite stable just with some surface ligand exchange between $PhCOO^-$ and $^iPrS^-$ such as those observed in paired species of **1b** and **1c**, **1 g** and **1 h**, and **1j** and **1k**. The 11 identified species existed in the solution, suggesting a typical coordination-disassociation equilibrium state, which may be broken by introducing some exotic stimuli, then producing some new species.

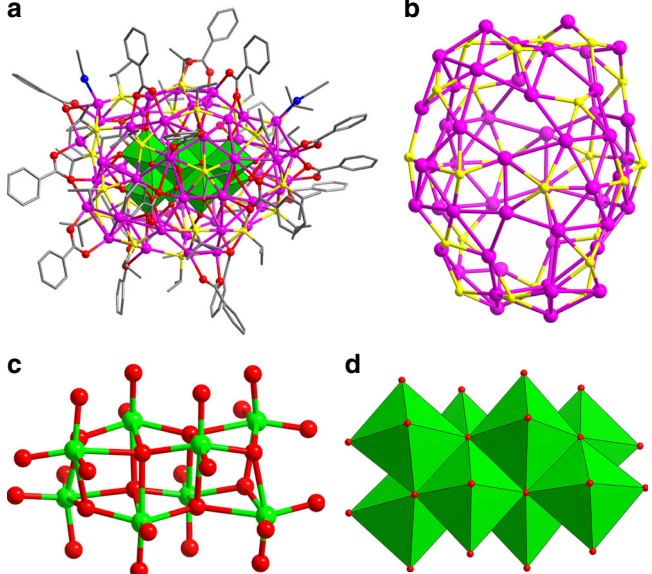

**Fig. 3** Single-crystal X-ray structure of **SD/Ag50**. **a** Molecular structure of **SD/Ag50** trapping a $Mo_8O_{28}^{8-}$ anion as template (Color legend: Ag: purple; S: yellow; Mo: green; O: red; C: gray; N: blue). **b** The ball-and-stick mode of $Ag_{50}S_{24}$ shell. The ball-and-stick (**c**) and polyhedral (**d**) modes showing the inner $Mo_8O_{28}^{8-}$ anion

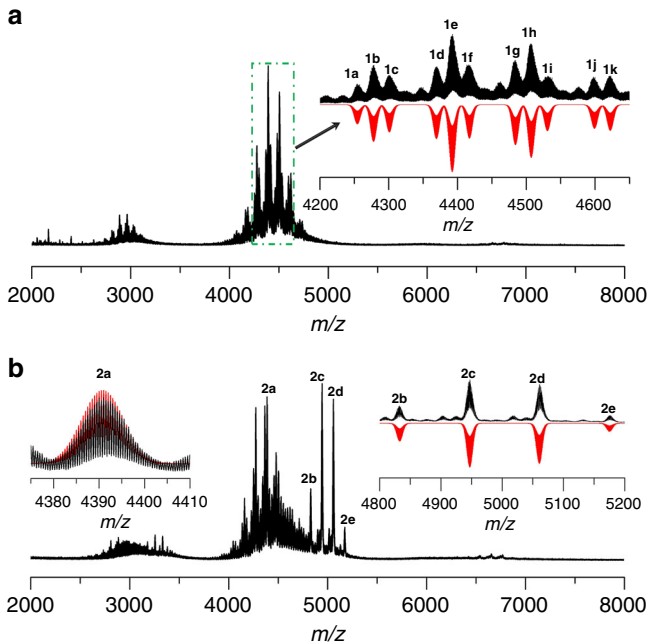

**Fig. 4** ESI-MS. **a** Positive ion mode ESI-MS of the crystals of **SD/Ag44** dissolved in dichloromethane. Inset: The zoom-in mass spectrum of experimental (black line) and simulated (red line) isotope patterns for each labeled species. **b** Positive ion mode ESI-MS of the crystals of **SD/Ag50** dissolved in dichloromethane. Insets: The zoom-in mass spectrum of experimental (black lines) and simulated (red lines) isotope patterns for each labeled species

More complicated ESI-MS was observed for **SD/Ag50** in its dichloromethane solution (Fig. 4b). Carefully analyzing the isotope distribution of the observed peaks in the $m/z$ range of 2000–8000, we found all these peaks were bivalent species with a parent ion peak of **SD/Ag50** detected at the $m/z$ 5175.19 (**2e**), corresponding to $[Mo_8O_{28}@Ag_{50}(^iPrS)_{24}(PhCOO)_{16}]^{2+}$ (Cal. 5175.23), which can be formed by losing all the acetonitrile molecules and two PhCOO$^-$ ligands from **SD/Ag50**. The neighbor peak centered at $m/z$ 5060.72 (**2d**) can be assigned to $[Mo_8O_{28}@Ag_{49}(^iPrS)_{24}(PhCOO)_{15}]^{2+}$ (Cal. 5060.76), corresponding to loss one PhCOOAg from **2e**. The most dominant peak (**2c**) was observed at $m/z$ 4947.25, assigned to $[Mo_8O_{28}@Ag_{47}(^iPrS)_{22}(PhCOO)_{15}(CH_2Cl_2)(H_2O)_3]^{2+}$ (Cal. 4947.32). The other identified species were shown in Supplementary Table 5. Of note, a series of peaks in the $m/z$ range of 4200–4700 are quite similar to those in ESI-MS of **SD/Ag44**. For example, the peak (**2a**) centered at 4391.01 can be ascribed to $[\Delta(^iPrS)_{25}(PhCOO)_9(CH_2Cl_2)(H_2O)_3]^{2+}$ (Cal. 4391.03). These results demonstrated (i) **SD/Ag50** can maintain its integrity in dichloromethane, and (ii) **SD/Ag50** can partially decompose to the closely related species to **SD/Ag44**.

**Transformation mechanism study.** How the transformation process happened and proceeded are intriguing issues in this system. To gain further insights into the transformation mechanism, we carried out a series of specific experiments including pH values measurements, comparable synthesis, and solution species tracking by ESI-MS during the transformation.

After synthesis of **SD/Ag44**, the pH of mother liquor is 10.45, then decreases to 8.20 upon adding PhCOOH, which is favorable to the further condensation reaction of molybdates[28]. To rule out

the $Mo_8O_{28}^{8-}$ being transformed from the residual molybdate species in mother liquor, we also used fresh crystals of **SD/Ag44** and clean $CH_3CN$ as solvent to do the transformation reaction and **SD/Ag50** can also be formed, which suggested the $Mo_8O_{28}^{8-}$ must be transformed from the interior $Mo_6O_{22}^{8-}$ in **SD/Ag44**. Meanwhile, this result also unambiguously indicated that the smaller-to-larger silver nanocluster conversion from **SD/Ag44** to **SD/Ag50** genuinely started from **SD/Ag44** instead of other silver precursors.

As we know, ESI-MS is a promising analytical tool that provides considerable speciation information in solution[29–32]. Cronin group have contributed largely to this field by using ESI-MS to detect new POM species and track transformation of POMs[33–35]. This technique was also profoundly used by Xie group to study growth mechanism of gold nanoclusters or reactivity[36–40]. However, using ESI-MS to study transformation or reactivity of silver clusters has remained a black box to date. To shed light on the details of transformation process from $Ag_{44}$ to $Ag_{50}$, we tracked the solution species evolution over the course of 240 min upon addition of PhCOOH (0.32 mmol, 39.1 mg) in the $CH_2Cl_2$ solution of $Ag_{44}$ cluster by time-dependent ESI-MS. In order to guarantee comparable intensity of signals for each species at different times, a generalized operation for the sampling and analysis was established and uniform instrument parameters were maintained. At scheduled time intervals, 500 μL aliquots of the reaction solution were extracted, and immediately infused (240 μL/h) to the mass spectroscopy without dilution for subsequent measurement. To set the baseline and finishing line for this monitoring process, we also incorporated the ESI-MS spectra of $Ag_{44}$ (black line) and $Ag_{50}$ (dark green line) in $CH_2Cl_2$ into Fig. 5, respectively, although in-source fragmentation of them under our chosen ESI source parameters have been studied in above section. All formulae of labeled species in the ESI-MS in this section are listed in Supplementary Table 6 and hereafter only inner core compositions are shown below for clarity. Zoom-in ESI-MS including the simulated isotope distributions of all these species are shown in Supplementary Fig. 5. As depicted in Fig. 5a (0 min), once adding PhCOOH, four doubly charged species **1l**-**1o** appeared in the lower $m/z$ range of 3600–4200 which are absent in original ESI-MS of $Ag_{44}$ solution. The assigned formulae for **1l**-**1o** are $[Mo_5O_{18}@Ag_{38}]$ (**1l**), $[Mo_5O_{18}@Ag_{39}]$ (**1m**), and $[Mo_5O_{18}@Ag_{40}]$ (**1n** and **1o**). In the higher $m/z$ range of 4600–5300, the other five larger divalent species (**1p**, **1q**, **1r**, **2d**, and **2e**) are observed and can be assigned to $[Mo_8O_{28}@Ag_{45}]$ (**1p**), $[Mo_8O_{28}@Ag_{46}]$ (**1q**), $[Mo_8O_{28}@Ag_{47}]$ (**1r**), $[Mo_8O_{28}@Ag_{49}]$ (**2d**), and $[Mo_8O_{28}@Ag_{50}]$ (**2e**). Of note, the **2e** species is the parent ion of **SD/Ag50**. All these intermediates are smaller or larger than original $Ag_{44}$ cluster and their formations are intensely dependent on the addition of PhCOOH.

The evolutions of species were further represented by the plots of signal intensity vs. time as shown in Fig. 5b. Upon adding PhCOOH, we observed four species (**1l**-**1o**) smaller than $Ag_{44}$, indicating the breakage of $Ag_{44}$ shell was initiated by PhCOOH. As time going on (5–240 mins), the signal intensities of $[Mo_5O_{18}@Ag_{38}]$ (**1l**), $[Mo_5O_{18}@Ag_{39}]$ (**1m**), and $[Mo_5O_{18}@Ag_{40}]$ (**1n** and **1o**) gradually decrease, whereas those of $[Mo_8O_{28}@Ag_{49}]$ (**2d**) and $[Mo_8O_{28}@Ag_{50}]$ (**2e**) become more intense, which suggested that the formation of $Ag_{50}$ cluster is at the expense of smaller intermediate species such as **1l**-**1o** at this stage. Moreover, we also noted the POM template encapsulated in silver shell undergo a condensation reaction from $Mo_6O_{22}^{8-}$ to $Mo_8O_{28}^{8-}$ and the excess oxygen coordination sites permit for the linkage of more silver atoms, thus forming the cluster larger than $Ag_{44}$.

Based on above observations, we speculated that the outer silver shell was attacked and partially broken by PhCOOH during

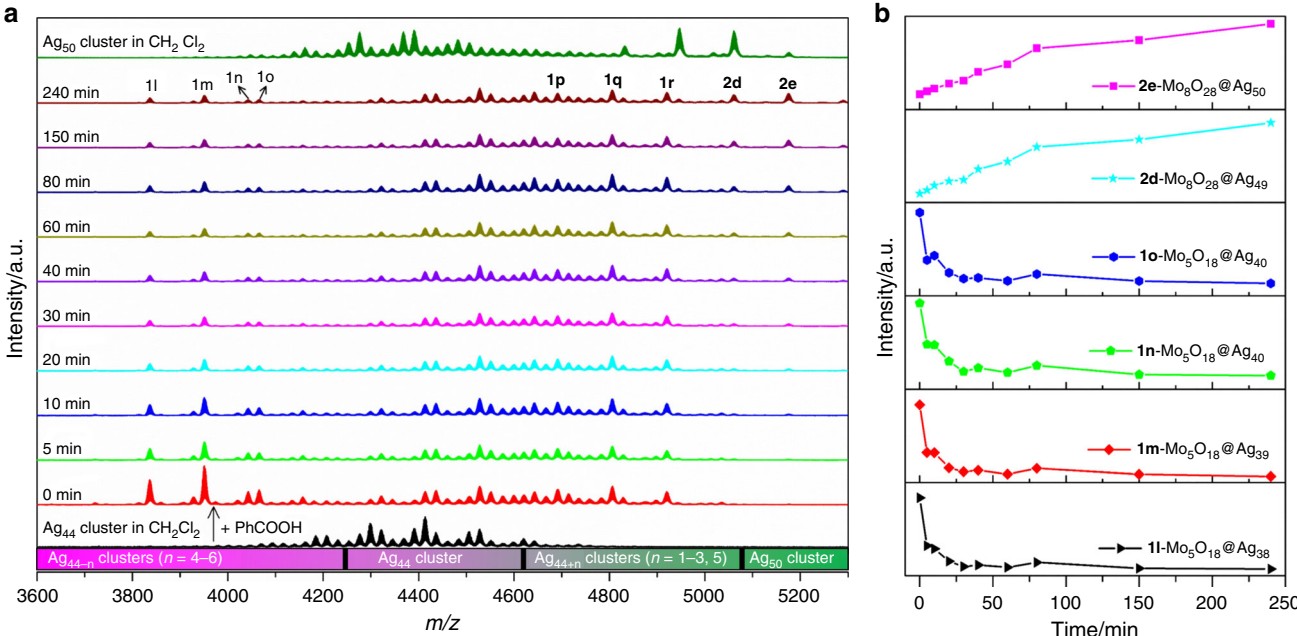

**Fig. 5** Time-course ESI-MS. **a** Time-course ESI-MS of transformation from Ag$_{44}$ to Ag$_{50}$ cluster induced by adding PhCOOH. The detailed formulae of labeled species are listed in Supplementary Table 6. **b** Plots of ESI-MS peak intensity of selected intermediate species vs time during the transformation process

the initial stage, which resulted in the breakage of Ag$_{44}$ shell in some extent, followed by the growth of inner Mo$_6$O$_{22}$$^{8-}$ anion to Mo$_8$O$_{28}$$^{8-}$. Due to the addition of PhCOOH, the solution becomes more acidity, thus larger Mo$_8$O$_{28}$$^{8-}$ was formed via condensation reaction. The decomposed fragments further reassemble around the new template to form final large Ag$_{50}$ cluster. Finally, the overall breakage-growth-reassembly (BGR) mechanism was established for this transformation process (Fig. 6).

**Universality of acid-induced transformation**. We also introduced different substituted benzoic acids such as 4-methylbenzoic acid (4-MePhCOOH) and 3-methylbenzoic acid (3-MePhCOOH) into above transformation process (Supplementary Fig. 1) to check the universality of transformation. Of note, the similar Ag$_{50}$ clusters can be also isolated after acid induction and characterized by SCXRD analysis, which showed their structures are quite similar with inner Mo$_8$O$_{28}$$^{8-}$ and outer Ag$_{50}$ shell but just with different substituted degree of benzoates on the surface. The details of molecular structures and crystallography tables were shown in Supplementary Figs. 6–8 and Supplementary Table 1, respectively. Their formulae were determined as [Mo$_8$O$_{28}$@Ag$_{50}$($^i$PrS)$_{24}$(4-MePhCOO)$_{14}$(PhCOO)$_4$(CH$_3$CN)$_4$] (**SD/Ag50a**), [Mo$_8$O$_{28}$@Ag$_{50}$($^i$PrS)$_{24}$(4-MePhCOO)$_{16.5}$(PhCOO)$_{1.5}$] (**SD/Ag50b**), and [Mo$_8$O$_{28}$@Ag$_{50}$($^i$PrS)$_{24}$(3-MePhCOO)$_{18}$(3-MePh-COOH)(CH$_3$CN)$_2$] (**SD/Ag50c**). The degree of substitution reaction between substituted benzoates and PhCOO$^-$ is intensely depended on the dose of substituted benzoates. For example, when increasing amount of 4-MePhCOOH from 0.3 to 0.6 mmol, the **SD/Ag50b** can be formed instead of **SD/Ag50a**. These results not only justified a universal acid-induced transformation fashion to synthesize larger sized silver clusters but also may bring new functionalities for silver clusters by ligand exchanges.

**UV-Vis absorption spectra and luminescence properties**. The solid state UV/Vis absorption spectra of **SD/Ag44**, **SD/Ag50**, and

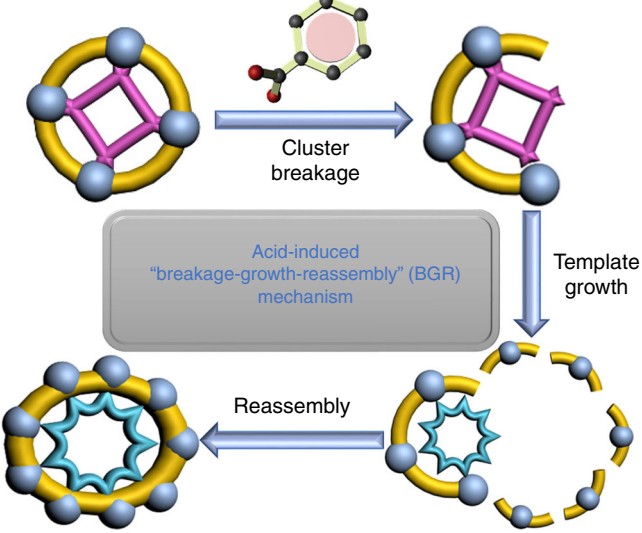

**Fig. 6** The proposed breakage-growth-reassembly mechanism for the transformation process from **SD/Ag44** to **SD/Ag50**

($^i$PrSAg)$_n$ precursor were measured at room temperature. As shown in Supplementary Fig. 19, **SD/Ag44** and **SD/Ag50** exhibit one intense absorption centered at 344 and 356 nm in the UV region. The UV absorption peaks can be attributed to the $n \rightarrow \pi^*$ transition of $^i$PrS$^-$, as similar observed in the spectrum of the precursor ($^i$PrSAg)$_n$. The HOMO–LUMO gaps were determined as 2.35, 2.33, and 2.52 eV for **SD/Ag44**, **SD/Ag50**, and ($^i$PrSAg)$_n$ precursor by using the transformed Kubelka–Munk plots, respectively (Supplementary Fig. 20), which are consistent with their yellow appearances.

We also checked the emission behaviors of **SD/Ag44** and **SD/Ag50** at both 298 and 77 K using hand-held UV light

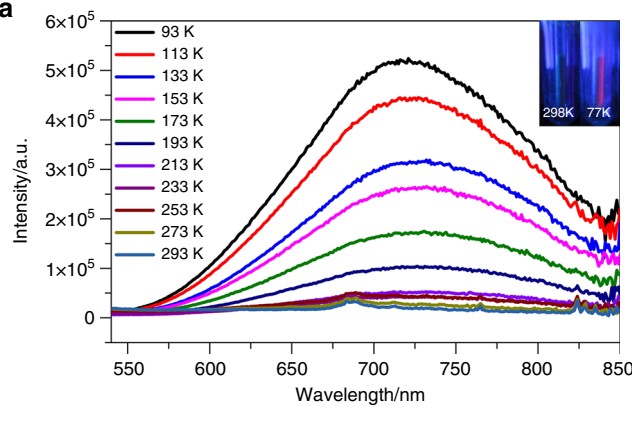

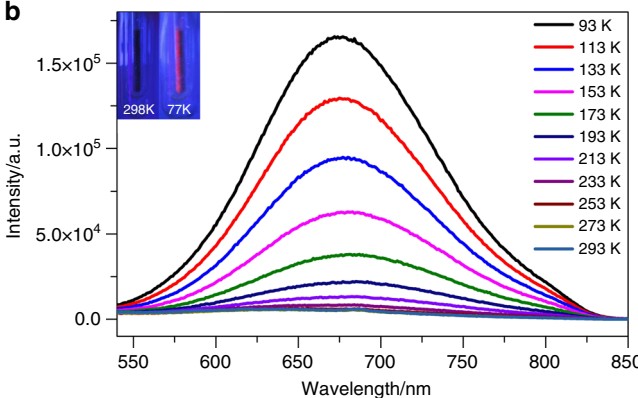

**Fig. 7** Luminescence properties of **SD/Ag44** and **SD/Ag50**. Varied-temperature emission spectra of **SD/Ag44** (**a**) and **SD/Ag50** (**b**) in solid state under the excitation of 463 nm, Insets: the photographs of **SD/Ag44** and **SD/Ag50** irradiated by hand-held UV light ($\lambda_{ex} = 365$ nm) at 298 and 77 K

($\lambda_{ex} = 365$ nm). Preliminarily, both **SD/Ag44** and **SD/Ag50** emit red luminescence at 77 K but become emission-silent at 298 K. The on-off phenomena of them are reversible between 298 and 77 K and can be directly observed by naked eyes (See the insets in Fig. 7). For detailed studies of such observations, varied-temperature luminescent spectra of **SD/Ag44** and **SD/Ag50** were measured from 93 to 293 K in the solid state. As shown in Fig. 7a, **SD/Ag44** emits near-infrared (NIR) light ($\lambda_{em} = 719$ nm) at 93 K and is almost non-emissive from 213–293 K. The maximum emission peak gradually red-shifts to 725 nm upon warming to 193 K along with obvious decrease of emission intensity. For **SD/Ag50**, its maximum emission peak red-shifts from 674 nm at 93 K to 686 nm at 233 K with gradually decayed emission intensity, after which no obvious emissions can be detected above 253 K. These NIR emissions at cryogenic temperature can be similarly attributed to the ligand-to-metal charge transfer (LMCT, charge transfer from S 3p to Ag 5 s) perturbed by Ag···Ag interaction[41,42]. The temperature-dependent emissions should be in the connection with the variations of molecule rigidity and argentophilicity under different temperatures[43,44]. The emission lifetimes of both **SD/Ag44** and **SD/Ag50** at 93 K fall in the microsecond scale (Supplementary Fig. 21), suggesting the phosphorescent triplet excitation state[45]. It is worth to noting that the emission intensities have good linearity correlation with respect to temperature in the low temperature regions (Supplementary Fig. 22). The linearity equations can be described as $I_{max} = -3977T + 876273$ (correlation coefficient = 0.985) and

$I_{max} = -1461T + 294270$ (correlation coefficient = 0.979) for **SD/Ag44** (93–213 K) and **SD/Ag50** (93–193 K), respectively.

## Discussion

In conclusion, we synthesized and characterized two thiolate/benzoate co-protected $Ag_{44}$ and $Ag_{50}$ clusters. The latter can be facilely synthesized by PhCOOH-induced transformation reaction, resulting in the enlargement of both inner anion template ($Mo_6O_{22}^{8-} \rightarrow Mo_8O_{28}^{8-}$) and outer silver shell ($Ag_{44} \rightarrow Ag_{50}$). The solution behaviors of $Ag_{44}$ and $Ag_{50}$ clusters were studied in details using ESI-MS technique. The overall breakage-growth-reassembly (BGR) transformation mechanism was also established based on the time-dependent ESI-MS, which revealed several small fragmented species ([$Mo_5O_{18}@Ag_{38}$]~ [$Mo_5O_{18}@Ag_{40}$]) and large intermediates ([$Mo_8O_{28}@Ag_{45}$]~ [$Mo_8O_{28}@Ag_{49}$]), as well as their abundance evolutions along with reaction going on. To our knowledge, this is the first example of the transformation of a smaller silver cluster into a larger one via anion template growth. This work not only revealed the important reactivity of silver clusters with acids but also provide a novel method for synthesizing larger silver clusters that are otherwise difficult to obtain.

## Methods

**Synthesis of ($^i$PrSAg)$_n$.** ($^i$PrSAg)$_n$ was prepared by the following reported procedure[43]. The solution of AgNO$_3$ (30 mmol, 5.1 g) in 75 mL acetonitrile was mixed with 100 mL ethanol containing $^i$PrSH (30 mmol, 2.8 mL) and 5 mL Et$_3$N under stirring for 3 h in the dark at room temperature, then the yellow powder of ($^i$PrSAg)$_n$ was isolated by filtration and washed with 10 mL ethanol and 20 mL ether, then dried in the ambient environment (yield: 97 %, based on AgNO$_3$).

**Synthesis of SD/Ag44.** Typically, the mixture of PhCOOAg (0.1 mmol, 22.9 mg), ($^i$PrSAg)$_n$ (0.05 mmol, 9.2 mg), and ($^n$Bu$_4$N)$_4$($\alpha$-Mo$_8$O$_{26}$) (0.0002 mmol, 4.2 mg) were dissolved in 5 mL acetonitrile then sealed into 25 mL Teflon-lined autoclave and heated at 65 °C under autogenous pressure for 2000 min. After cooling to room temperature, yellow crystals were isolated with a yield of 45 % (based on ($^i$PrSAg)$_n$). Elemental analyses calc. (found) for **SD/Ag44** (C$_{180}$H$_{232}$Ag$_{44}$Mo$_6$N$_4$O$_{54}$S$_{20}$): C, 23.30 (23.18); H, 2.52 (2.47); N 0.60 (0.55) %. Selected IR peaks (cm$^{-1}$): 3682 (w), 2961 (m), 1589 (w), 1523 (m), 1454 (w), 1371 (s), 1241 (w), 1149 (w), 1032 (s), 1014 (m), 811 (w), 715 (m), 675 (m), 588 (w).

**Synthesis of SD/Ag50.** Another portion of PhCOOH (0.32 mmol, 39.1 mg) was added into the mother liquor after the synthesis of **SD/Ag44** but without removing crystals of **SD/Ag44**, then this mixture was again sealed into 25 mL Teflon-lined autoclave and heated at 65 °C under autogenous pressure for 2000 min. After cooling to room temperature, yellow crystals were isolated with a yield of 50% (based on ($^i$PrSAg)$_n$). Elemental analyses calc. (found) for **SD/Ag50** (C$_{210}$H$_{276}$Ag$_{50}$Mo$_8$N$_6$O$_{64}$S$_{24}$): C, 23.09 (23.18); H, 2.57 (2.48); N 0.78 (0.70) %. Selected IR peaks (cm$^{-1}$): 2955 (w), 2907 (w), 1589 (w), 1528 (s), 1454 (w), 1371 (s), 1241 (m), 1144 (w), 1046 (m), 910 (w), 880 (m), 830 (m), 800 (m), 714 (s), 671 (m), 635 (m), 610 (m).

**Synthesis of SD/Ag50a.** The synthesis conditions were similar to those described for **SD/Ag50** except using 4-MePhCOOH (0.33 mmol, 44.9 mg) instead, yellow crystals were isolated with a yield of 36 % (based on ($^i$PrSAg)$_n$). Elemental analyses calc. (found) for **SD/Ag50a** (C$_{220}$H$_{298}$Ag$_{50}$Mo$_8$N$_4$O$_{64}$S$_{24}$): C, 24.12 (23.97); H, 2.74 (2.58); N 0.51 (0.46) %. Selected IR peaks (cm$^{-1}$): 3670 (w), 2949 (w), 2912 (w), 1584 (w), 1520 (m), 1453 (w), 1368 (s), 1242 (m), 1169 (m), 1148 (m), 1052 (m), 916 (m), 878 (m), 831 (m), 796 (s), 759 (s), 712 (m), 632 (s), 606 (s).

**Synthesis of SD/Ag50b.** The synthesis conditions were similar to those described for **SD/Ag50** except using 4-MePhCOOH (0.57 mmol, 77.6 mg) instead, yellow crystals were isolated with a yield of 30 % (based on ($^i$PrSAg)$_n$). Elemental analyses calc. (found) for **SD/Ag50b** (C$_{214.5}$H$_{291}$Ag$_{50}$Mo$_8$O$_{64}$S$_{24}$): C, 23.80 (23.59); H, 2.71 (2.58) %. Selected IR peaks (cm$^{-1}$): 3674 (w), 2971 (w), 2905 (w), 1584 (w), 1524 (m), 1453 (w), 1368 (s), 1237 (m), 1173 (w), 1149 (w), 1047 (s), 911 (m), 877 (s), 834 (m), 798 (m), 763 (s), 712 (m), 632 (s), 605 (s).

**Synthesis of SD/Ag50c.** The synthesis conditions were similar to those described for **SD/Ag50** except using 3-MePhCOOH (0.57 mmol, 77.6 mg), yellow crystals were isolated with a yield of 33 % (based on ($^i$PrSAg)$_n$). Elemental analyses calc. (found) for **SD/Ag50c** (C$_{234}$H$_{317}$Ag$_{50}$Mo$_8$N$_5$O$_{66}$S$_{24}$): C, 25.12 (24.98); H, 2.86

(2.74); N 0.63 (0.57) %. Selected IR peaks (cm$^{-1}$): 3673 (w), 2955 (w), 2912 (w), 1592 (w), 1528 (m), 1450 (w), 1368 (s), 1236 (m), 1149 (m), 1078 (m), 1049 (m), 914 (m), 875 (m), 780 (s), 714 (m), 671 (s), 631 (s), 601 (s).

## Data availability

The X-ray crystallographic coordinates for structures reported in this article have been deposited at the Cambridge Crystallographic Data Centre, under deposition number CCDC: 1837115–1837119 for **SD/Ag44**, **SD/Ag50**, and **SD/Ag50a-SD/Ag50c**. These data can be obtained free of charge from the Cambridge Crystallographic Data Centre via www.ccdc.cam.ac.uk/data_request/cif.

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

## Acknowledgements

This work was financially supported by the National Natural Science Foundation of China (Grant Nos. 21822107, 21571115, 21827801, and 21701133), the Natural Science Foundation of Shandong Province (Nos. JQ201803 and ZR2017MB061), the Qilu Youth Scholar Funding of Shandong University and the Fundamental Research Funds of Shandong University (104.205.2.5). H.-F.S. thanks the President Research Funds from Xiamen University (20720170100).The authors also give special thanks to Dr. Zhao-Zhen Cao from Shandong University for her help with the varied-temperature luminescence spectra measurements.

## Author contributions

Original idea was conceived by D.S., experiments and data analyses were performed by Z.W. and D.S., ESI-MS data were collected by H.-F.S., structure characterization was

performed by Z.W. and D.S., manuscript was drafted by D.S., Z.W., C.-H.T. and L.-S.Z. All authors have given approval to the manuscript.

## Additional information

**Competing interests:** The authors declare no competing interests.

