## [Peer Review File · Nature Communications]

Reviewers' comments:

Reviewer #1 (Remarks to the Author):

The transformation from one atomically precise metal nanocluster to another is one of the ways to prepare the atomically precise metal nanoclusters, especially for those not easy to obtain via direct synthesis. Understanding the transformation process is significant for the design of this process, and in-depth understanding the properties of these metal nanoclusters can be also gained. In this manuscript, the authors presented their investigation on a synergetic core-shell transformation from [Mo₆O₂₂@Ag₄₄] (SD/Ag₄₄) to [Mo₈O₂₈@Ag₅₀] (SD/Ag₅₀). Based on the intermediates they found from ESI-MS spectra, a "breakage-growth-reassembly" mechanism was proposed. This work revealed the mechanism of the transformation from a small silver nanocluster to a larger one (realising the total synthesis of this metal nanocluster species), which was not known before but of great significance. The reactivity of the silver nanoclusters, the size transformation method, and the revealed mechanism will be inspiring for researchers working in the field. I believe this study will be of interest to heterogeneous readers from communities of noble metal chemistry, cluster chemistry, inorganic chemistry and materials chemistry. The manuscript is well-written, and I would like to suggest the acceptance of this paper after the authors have addressed the following minor issues.

1. Is it possible to determine the conversion of the size transformation reaction from SD/Ag₄₄ to SD/Ag₅₀? Maybe some physical chemistry insights (e.g. relative stability) can be gained with the yield of SD/Ag₅₀ in this reaction.
2. The authors mentioned above 50% yield of SD/Ag₅₀ of using the transformation synthesis route in the manuscript (page 7), but the yield they presented is only 30% in the method section. The authors need to include the basis of yield calculation to avoid confusion.
3. If the conversion from SD/Ag₄₄ to SD/Ag₅₀ is not 100%, it could be possible that in figure 4b, the peaks in the m/z range between 4200-4700 are from SD/Ag₄₄ instead of from the decomposition of SD/Ag₅₀.
4. The plots in figure 5b can be simply linked together with lines instead of fitting them with a curve. The fitting of the plots should come with some theoretical models predicting the decay of the intermediates with time.
5. The optical properties of SD/Ag₄₄ is currently somehow not related to the focus of this study (the transformation reaction). It will be better to include the optical properties of SD/Ag₅₀ for comparison so that readers can understand the property change after the size transformation reaction.

Reviewer #2 (Remarks to the Author):

In this work, Sun et. al., made a breakthrough in silver cluster chemistry where benzoate-induced core-shell synergetic transformation from [Mo₆O₂₂@Ag₄₄] to [Mo₈O₂₈@Ag₅₀] was realized, and more important, the fundamental insight on the conversion mechanism from [Mo₆O₂₂@Ag₄₄] to [Mo₈O₂₈@Ag₅₀] was unambiguously established as so-called "Breakage-Growth-Reassembly" based on the skillful ESI-MS analysis. The descriptions of the structure and conversion process are very well done, clear, and with detail. This topic is a very important one, as the better understanding could lead to a more rational synthesis of silver clusters with a controlled fashion, which should be very appealing and of general interest to many researchers. Based on the conversion mechanism, they also extended

such silver cluster conversion reaction to a larger extent using different substituted benzoic acids, thus giving a universal route to get larger silver cluster from the smaller one. I believe this research will deep our understanding on the synthesis and reaction of core-shell silver nanoclusters under the external chemical stimuli. Overall this is an excellent and valuable contributions, thus I recommend the publication of this work in Nature Communications after the authors address the following minor points:

- (1) ESI-MS have confirmed the solution stabilities of the silver nanoclusers, but as we know, many silver compounds are not thermalstable, so the stabilities of silver nanocluster crystals in solid state need to be studied.
- (2) Molybdates have many kinds of forms such as common PMo_{12} and Mo_7O_{24} in POM chemistry. Have authors found the Mo-sources influence the assembly results?
- (3) Are all silver atoms in Ag_{44} and Ag_{50}^{+1} ? Is there some possible to few reduced $\text{Ag}(0)$ in this assembly system?
- (4) The luminescence research is a little bit preliminary only with some varied temperature spectra and emission lifetime, which is not sufficient for such reputable journal. The luminescence quantum yield should be measured.
- (5) As claimed by authors, when adding 0.32 mmol PhCOOH into SD/Ag_{44} system, it was converted to SD/Ag_{50} . Thus if adding more benzoic acid into SD/Ag_{44} system, whether the higher nuclearity silver cluster trapping larger POM template will be isolated?
- (6) In several places in the manuscript, there is some awkward phrasing. You may wish to go through and wordsmith the document some more to eliminate these.

Reviewer 1:

The transformation from one atomically precise metal nanocluster to another is one of the ways to prepare the atomically precise metal nanoclusters, especially for those not easy to obtain via direct synthesis. Understanding the transformation process is significant for the design of this process, and in-depth understanding the properties of these metal nanoclusters can be also gained. In this manuscript, the authors presented their investigation on a synergetic core-shell transformation from [Mo₆O₂₂@Ag₄₄] (SD/Ag₄₄) to [Mo₈O₂₈@Ag₅₀] (SD/Ag₅₀). Based on the intermediates they found from ESI-MS spectra, a “breakage-growth-reassembly” mechanism was proposed. This work revealed the mechanism of the transformation from a small silver nanocluster to a larger one (realising the total synthesis of this metal nanocluster species), which was not known before but of great significance. The reactivity of the silver nanoclusters, the size transformation method, and the revealed mechanism will be inspiring for researchers working in the field. I believe this study will be of interest to heterogeneous readers from communities of noble metal chemistry, cluster chemistry, inorganic chemistry and materials chemistry. The manuscript is well-written, and I would like to suggest the acceptance of this paper after the authors have addressed the following minor issues.

Response: We are pleased and excited by above positive comments on the novelty and significance of our study.

(1) Is it possible to determine the conversion of the size transformation reaction from SD/Ag₄₄ to SD/Ag₅₀? Maybe some physical chemistry insights (e.g. relative stability) can be gained with the yield of SD/Ag₅₀ in this reaction.

Response: Thanks for your insightful suggestions.

(1) This is a really good comment about the origin of conversion reaction between silver clusters. We are sorry for the confusion caused by less-detailed interpretation of compared conversion experiments in the

original submission. We initially observed the crystals of SD/Ag50 formed after the second step reaction, before which another portion of PhCOOH was added into the reaction mother liquor containing the crystals of SD/Ag44. The conversion formation of SD/Ag50 was confirmed by single-crystal X-ray diffraction (SCXRD) analysis. More important, to rule out the SD/Ag50 being transformed from the residual silver species in mother liquor, we also used fresh crystals of SD/Ag44 and clean CH₃CN as solvent to do the transformation reaction and SD/Ag50 can also be formed, which unambiguously indicated that the smaller-to-larger silver nanocluster conversion from SD/Ag44 to SD/Ag50 genuinely started from SD/Ag44 instead of other silver precursors. For this case, we added one sentence in Page 9 as: "Meanwhile, this result also unambiguously indicated that the smaller-to-larger silver nanocluster conversion from SD/Ag44 to SD/Ag50 genuinely started from SD/Ag44 instead of other silver precursors."

(2) We fully agree with his/her insightful suggestion. From some physical chemistry viewpoints, this transformation reaction should be a thermodynamics controlled reaction, so SD/Ag50 in this system should be more stable product than SD/Ag44. For this case, we also added one sentence in Page 7 as: "Moreover, this transformation reaction should be a thermodynamics controlled reaction, so SD/Ag50 should be more stable than SD/Ag44."

(2) The authors mentioned above 50% yield of SD/Ag50 of using the transformation synthesis route in the manuscript (page 7), but the yield they presented is only 30% in the method section. The authors need to include the basis of yield calculation to avoid confusion.

Response: Thanks for the important reminder about the calculation of yield of SD/Ag50. We are sorry for the confusion caused by missing the basis of yield calculation in the original synthesis section. We carefully rechecked the yield calculation basis that should be the insufficient reactant, here is

(iPrSAg)_n. So we corrected the yield reported in synthesis section. We also added all yield calculation basis for all 6 compounds including one silver precursor and five silver nanoclusters.

(3) If the conversion from SD/Ag44 to SD/Ag50 is not 100%, it could be possible that in figure 4b, the peaks in the m/z range between 4200-4700 are from SD/Ag44 instead of from the decomposition of SD/Ag50.

Response: Thanks for your constructive suggestion. We carefully reconsidered the species origin in the m/z range of 4200-4700 in Figure 4b. Firstly, we are sure that the crystalline sample of SD/Ag50 is pure as confirmed by compared powder X-ray diffraction (PXRD) patterns (Supplementary Figure 15). Secondly, the ESI-MS shown in Figure 4b was obtained using the dichloromethane solution of pure SD/Ag50 crystals that were collected by manually picking one by one, so the peaks in the m/z range between 4200-4700 must be from the decomposition of SD/Ag50 instead of SD/Ag44. I hope these explanations can well address the concern from this reviewer.

(4) The plots in figure 5b can be simply linked together with lines instead of fitting them with a curve. The fitting of the plots should come with some theoretical models predicting the decay of the intermediates with time.

Response: Thanks for your insightful suggestion. We have corrected the plots in Figure 5b with lines linked together.

(5) The optical properties of SD/Ag44 is currently somehow not related to the focus of this study (the transformation reaction). It will be better to include the optical properties of SD/Ag50 for comparison so that readers can understand the property change after the size transformation reaction.

Response: Thanks for your constructive suggestions. Yes, the focus of this study is the silver cluster transformation reaction instead of the optical properties of these clusters. Anyway, as suggested by this reviewer, we rechecked the emission behavior of SD/Ag50 at both room temperature and 77 K during the revision stage (see figure below). Under the UV excitation,

SD/Ag50 is also emission-silent at room temperature and shows similar emission turn-on behavior at low temperature. We collected its varied-temperature luminescent spectra from 293-93 K and the related results and discussions were added into main text as: *"We also checked the emission behaviors of SD/Ag44 and SD/Ag50 at both 298 and 77 K using hand-held UV light ($\lambda_{ex} = 365$ nm). Preliminarily, both SD/Ag44 and SD/Ag50 emit red luminescence at 77 K but become emission-silent at 298 K. The on-off phenomena of them are reversible between 298 and 77 K and can be directly observed by naked eyes (See the insets in Fig. 7). For detailed studies of such observations, varied-temperature luminescent spectra of SD/Ag44 and SD/Ag50 were measured from 93 to 293 K in the solid state. As shown in Fig. 7a, SD/Ag44 emits near-infrared (NIR) light ($\lambda_{em} = 719$ nm) at 93 K and is almost non-emissive from 213-293 K. The maximum emission peak gradually red-shifts to 725 nm upon warming to 193 K along with obvious decrease of emission intensity. For SD/Ag50, its maximum emission peak red-shifts from 674 nm at 93 K to 686 nm at 233 K with gradually decayed emission intensity, after which no obvious emissions can be detected above 253 K. These NIR emissions at cryogenic temperature can be similarly attributed to the ligand-to-metal charge transfer (LMCT, charge transfer from S 3p to Ag 5s) perturbed by Ag...Ag interaction.^{41,42} The temperature-dependent emissions should be in the connection with the variations of molecule rigidity and argentophilicity under different temperatures.⁴³ The emission lifetimes of both SD/Ag44 and SD/Ag50 at 93 K fall in the microsecond scale (Supplementary Fig. 21), suggesting the phosphorescent triplet excitation state.⁴⁴ It is worth to noting that the emission intensities have good linearity correlation with respect to temperature in the low temperature regions (Supplementary Fig. 22). The linearity equations can be described as $I_{max} = -3977T + 876273$ (correlation coefficient = 0.985) and $I_{max} = -1461T + 294270$ (correlation coefficient = 0.979) for SD/Ag44 (93-213 K) and SD/Ag50 (93-193 K), respectively."*

Reviewer 2:

In this work, Sun et. al., made a breakthrough in silver cluster chemistry where benzoate-induced core-shell synergetic transformation from [Mo₆O₂₂@Ag₄₄] to [Mo₈O₂₈@Ag₅₀] was realized, and more important, the fundamental insight on the conversion mechanism from [Mo₆O₂₂@Ag₄₄] to [Mo₈O₂₈@Ag₅₀] was unambiguously established as so-called “Breakage-Growth-Reassembly” based on the skillful ESI-MS analysis. The descriptions of the structure and conversion process are very well done, clear, and with detail. This topic is a very important one, as the better understanding could lead to a more rational synthesis of silver clusters with a controlled fashion, which should be very appealing and of general interest to many researchers. Based on the conversion mechanism, they also extended such silver cluster conversion reaction to a larger extent using different substituted benzoic acids, thus giving a universal route to get larger silver cluster from the smaller one. I believe this research will deep our understanding on the synthesis and reaction of core-shell silver nanoclusters under the external chemical stimuli. Overall this is an excellent and valuable contributions, thus I recommend the publication of this work in Nature Communications after the authors address the following minor points:

Response: Thanks for these positive comments on the novelty and significance of our study. We share the reviewer’s view that our findings deepen our understanding on the synthesis and reaction of core-shell silver nanoclusters under the external chemical stimuli.

(1) ESI-MS have confirmed the solution stabilities of the silver nanoclusers, but as we know, many silver compounds are not thermalstable, so the stabilities of silver nanocluster crystals in solid state need to be studied.

Response: Thank you for this insightful comment. We identified the thermal stabilities of SD/Ag₄₄ and SD/Ag₅₀ by Thermogravimetric Analysis (TGA). The weight loss curves were added in Supplementary Figure S23 and also shown below. SD/Ag₄₄ and SD/Ag₅₀ showed similar

stability in the temperature range of 20-800 °C, and the starting decomposition temperature is near to 160 °C.

(2) Molybdates have many kinds of forms such as common PMo_{12} and Mo_7O_{24} in POM chemistry. Have authors found the Mo-sources influence the assembly results?

Response: Thanks for your constructive suggestion. We also tried to use other molybdenum sources as anion templates, such as $\text{Na}_2\text{MoO}_4 \cdot 2\text{H}_2\text{O}$, $(\text{nBu}_4\text{N})_2\text{Mo}_6\text{O}_{19}$ and $(\text{NH}_4)_6\text{Mo}_7\text{O}_{24} \cdot 4\text{H}_2\text{O}$, in this assembly system, however, no any crystalline products can be isolated, which suggested the molybdenum sources also played an important role in the formation of Ag_{44} and Ag_{50} clusters.

(3) Are all silver atoms in Ag_{44} and Ag_{50} +1? Is there some possible to few reduced $\text{Ag}(0)$ in this assembly system?

Response: Thank you for this insightful comment. Yes, the oxidation state of all silver atoms in Ag_{44} and Ag_{50} is +1 based on the charge neutrality consideration and reaction conditions. There are no any reductive reactant or solvent in this assembly system, so no any $\text{Ag}(0)$ species appeared in this

assembly system, which was also consistent with the observation that no silver mirror appeared on the wall of reaction vessel.

If a certain amount of reductive solvent is added to the system, such as DMF solvent, the silver nanocluster containing Ag(0) atom may be formed as seen in our recent work (Wang, Z. *et al.* Trapping an octahedral Ag₆ kernel in a seven-fold symmetric Ag₅₆ nanowheel. *Nat Commun* 9, 2094 (2018).), which showed that DMF can be used as a weak reductant to reduce silver(I) to nanocluster containing Ag(0). In this work, we only used redox inert acetonitrile as solvent, so it is impossible to get Ag(0) species.

(4) The luminescence research is a little bit preliminary only with some varied temperature spectra and emission lifetime, which is not sufficient for such reputable journal. The luminescence quantum yield should be measured.

Response: Thank you for this constructive comment. As mentioned from reviewer 1, the optical property is currently not the focus of this study (the transformation reaction). Furthermore, SD/Ag44 only emits at cryogenic temperature, however, it is emission-silent at room temperature. While our fluorescence spectrophotometer can only measure the luminescence quantum yield at room temperature, so due to the limitation of our fluorescence spectrophotometer (Edinburgh spectrofluorimeter (F920S)), we cannot provide the luminescence quantum yield at low temperature. Moreover, we also supplemented the luminescence of SD/Ag50 into main text to enhance the overall quality of this work.

(5) As claimed by authors, when adding 0.32 mmol PhCOOH into SD/Ag44 system, it was converted to SD/Ag50. Thus if adding more benzoic acid into SD/Ag44 system, whether the higher nuclearity silver cluster trapping larger POM template will be isolated?

Response: Thanks for your constructive suggestion. We have added a series of amounts of PhCOOH from 0.3 to 1 mmol into SD/Ag44 conversion reaction system. To our disappointment, there is no higher nuclearity silver cluster trapping larger POM template isolated except for SD/Ag50.

(6) In several places in the manuscript, there is some awkward phrasing. You may wish to go through and wordsmith the document some more to eliminate these.

Response: Thanks for your constructive suggestion. We carefully revised several typos and awkward phrasings for easily reading. The further English polishing was also performed.

REVIEWERS' COMMENTS:

Reviewer #1 (Remarks to the Author):

The authors have addressed all my concerns. I am very enthusiastic to recommend the acceptance of this nice paper.

Reviewer #2 (Remarks to the Author):

The points raised in the previous round of review have been satisfactorily addressed. The revised manuscript is suitable for publication.

Reviewer #1 (Remarks to the Author):

The authors have addressed all my concerns. I am very enthusiastic to recommend the acceptance of this nice paper.

Response: Thank you very much. We appreciate your recommendation for publication of our work in Nature Communications.

Reviewer #2 (Remarks to the Author):

The points raised in the previous round of review have been satisfactorily addressed. The revised manuscript is suitable for publication.

Response: Thank you very much. We appreciate your recommendation for publication of our work in Nature Communications.